# γδ T cell receptor recognition of CD1d in a lipid-independent manner

Michael T. Rice [1], Sachith D. Gunasinghe[1,3], Chhon Ling Sok [1], Mengqi Pan [1], Chan-Sien Lay[1], Benjamin S. Gully [1,3] ✉ & Jamie Rossjohn [1,2] ✉

The monomorphic antigen-presenting molecule CD1d presents lipid antigens to both αβ and γδ T cells. While type I natural killer T cells (NKT) display exquisite specificity for CD1d presenting α-galactosylceramide (α-GalCer), the extent of lipid specificity exhibited by CD1d-restricted γδ T cells remains unclear. Here, we demonstrate that human γδ T cell receptors (TCRs) can recognise CD1d in either a lipid-dependent or lipid-independent manner with weak to moderate affinity. Using small-angle X-Ray scattering, we find that γδ TCR-CD1d binding modality is conserved across distinct CD1d-restricted TCRs. In functional assays, CD1d γδ TCR affinity was a poor predictor of γδ T cell line activation. Moreover, CD1d presenting endogenous lipids was sufficient to stimulate T cell activation and induce γδ TCR-CD3 clustering and phosphorylation in a dose-dependent manner. Elongation of the γδ TCR-CD3 complex by the inclusion of the Cγ2 and Cγ3 -encoded constant domains perturbed cellular activation whilst maintaining the ability to form functional γδ TCR clusters. The crystal structure of a Vδ1 γδ⁺ TCR-CD1d complex showed that the γδ TCR sat atop of the CD1d antigen-binding cleft but made no contacts with the presented lipid antigen. These findings provide a molecular basis for lipid-independent CD1d recognition by γδ TCRs.

The human adaptive immune compartment comprises two lineages of T cells expressing heterodimeric T cell receptors (TCR), αβ and γδ. γδ T cells are a diverse population of unconventional lymphocytes that display broad effector functions of both adaptive and innate immune cells[1]. Unconventional αβ T cells, such as mucosal-associated invariant T cells and natural killer T cells (NKT), which recognise MR1 and CD1d, respectively, have been described as innate-like due to their restricted TCR usage and rapid onset of effector functions[2]. γδ T cells can display similar innate-like characteristics, exhibiting limited TCR diversity and eliciting effector functions via innate receptors[3]. Conversely, clonal expansion of γδ T cell clones in Merkel cell carcinoma, as well as malarial and cytomegalovirus infections, has been observed, suggesting specific development of adaptive γδ T cells[4–6].

γδ T cell subsets include Vδ1⁺ and Vδ2⁺ compartments that are highly abundant within peripheral tissues and peripheral blood, respectively. Vδ1⁺ γδ T cells are enriched at epithelial sites such as the gut, liver or lungs (5–10%) and present at low frequency within peripheral blood (< 1%)[7–9]. Vδ1⁺ γδ T cells are an adaptive-like T cell population, with peripheral blood and tissue-resident Vδ1⁺ γδ T cells undergoing clonal expansion from an initially diverse pool of TCR sequences, accompanied by a phenotypic change[7,10]. Human and mouse intestinal mucosa contain γδ intraepithelial lymphocytes (IELs) that rapidly mobilise upon infection to maintain barrier integrity[11,12]. However, the precise mechanism underpinning their function is complex, as their infection-sensing and cytotoxic effector functions can occur in either a TCR-dependent or -independent manner[11,12]. It is clear, however, that there is a crosstalk between gut microbiota and γδ

¹Infection and Immunity Program and Department of Biochemistry and Molecular Biology, Biomedicine Discovery Institute, Monash University, Clayton, Victoria, Australia. ²Institute of Infection and Immunity, Cardiff University School of Medicine, Heath Park, Cardiff, United Kingdom. ³Present address: Olivia Newton-John Cancer Research Institute, Heidelberg, VIC, Australia. School of Cancer Medicine, La Trobe University, Heidelberg, VIC, Australia. ✉e-mail: Ben.gully@monash.edu; Jamie.rossjohn@monash.edu

T cells that serves to regulate cellular function[11,13]. Li and colleagues demonstrated that disruption of the microbiota by antibiotic treatment resulted in a significant reduction of the number of IL-17A-producing hepatic γδ T cells[13]. Retention of hepatic IL-17A[+] γδ T cells was dependent upon CD1d lipid antigen presentation, as CD1d-deficient mice had a reduced IL-17A[+] T cell population akin to antibiotic-treated mice[13].

In humans, CD1d has been shown to present endogenous and exogenously-derived antigens to both peripheral blood and IEL Vδ1[+] and Vδ3[+] γδ T cells[14–19]. CD1d and γδ T cells are both enriched in barrier sites, such as the liver, indicating TCR recognition of CD1d is potentially involved in tissue γδ T cell retention and cellular activation in humans[8,20]. Human CD1d-reactive γδ T cells are primarily CD4[-]CD8[-], whilst the majority of type I αβ NKT cells are CD4[+] and, unlike the lipid-specific type I αβ NKT cells, peripheral blood mononuclear cells (PBMC)-derived γδ T cells from healthy donors recognise CD1d presenting an array of lipid antigens[19]. Structures of two Vδ1[+] γδ TCRs bound to CD1d revealed that γδ TCRs can make direct contacts to the presented lipid antigens[18,19]. Direct recognition of the presented antigen directly contrasts the recently resolved antigen-independent binding by MR1 and CD1a-reactive γδ TCRs[21–23]. Similarly, γδ IELs produced more TNF-α in response to CD1d presenting endogenously derived lipids, rather than CD1d-sulfatide, raising questions on the requirement of lipid engagement to mediate γδ T cell effector functions[18]. Previous tetramer staining and activation experiments on primary Vδ1[+] γδ[+] CD1d-α-GalCer[+] cells suggested donor-specific patterns of lipid-independent and -dependent cell staining and CD1d-dependent activation[19].

Here, we characterise a panel of these previously identified CD1d-α-GalCer reactive Vδ1[+] T cells to determine the extent to which Vδ1[+] CD1d binding is lipid-dependant[19]. We report that γδ TCR recognition of CD1d is either unaltered or enhanced upon lipid engagement. Further, we determine the structure of a γδ TCR recognising CD1d that binds at a site distal to the presented lipid antigen. We provide the molecular basis of CD1d lipid-independent recognition by a γδ TCR.

## Results

### Affinity of Vδ1[+] γδ TCRs for CD1d presenting lipid antigens

To offer broad insight into Vδ1[+] γδ TCR recognition of CD1d, we used a panel of previously identified CD1d-reactive γδ TCRs, including diverse gene elements, Vδ1Vγ5[+] (TCR2), Vδ1Vγ2[+] (TCR3), Vδ1Vγ9[+] (TCR6), Vδ1Vγ5[+] (TCR7), Vδ1Vγ5[+] (TCR8) (Supplementary Table 1)[19]. We recombinantly expressed and purified these γδ TCRs alongside the control 9C2 γδ TCR and the NKT.15 αβ TCR[19,24]. We next conducted surface plasmon resonance (SPR) with immobilised CD1d loaded with α-GalCer (CD1d-α-GalCer), endogenously loaded lipids (CD1d-'endo') and CD1b-'endo'. None of the γδ TCRs tested here recognised CD1b-'endo' and specifically bound CD1d via two divergent patterns of specificity (Supplementary Fig. 1). The γδ TCRs 2, 3 and 6 bound to CD1d-'endo' ($K_D = 3.34 \pm 0.24\ \mu M$, $K_D = 6.63 \pm 2.09\ \mu M$, $K_D = 5.83 \pm 1.19\ \mu M$) and CD1d-α-GalCer ($K_D = 3.70 \pm 0.33\ \mu M$, $K_D = 3.09 \pm 0.74\ \mu M$, $K_D = 5.08 \pm 0.98\ \mu M$) with moderate affinity comparable to other γδ TCR-MHC-I like interactions[18,21–23,25](Fig. 1A). The lipid-independent recognition of CD1d by γδ TCR 2, 3 and 6 contrasted with the NKT.15 αβ TCR, which displayed high-affinity binding to CD1d-α-GalCer ($K_D = 0.44 \pm 0.04\ \mu M$) and weak binding to CD1d-'endo'. The apparent lipid cross-reactive Vδ1[+] γδ TCR CD1d binding observed here has also been observed for CD1a and CD1b specific γδ TCRs that displayed 'lipid-agnostic' binding[21,25].

Comparatively, γδ TCRs 7 and 8, weakly recognised CD1d-'endo'($K_D = 36.89 \pm 8.34\ \mu M$, $K_D = 62.74 \pm 9.51\ \mu M$), yet bound CD1d-α-GalCer with higher affinity ($K_D = 13.84 \pm 2.62\ \mu M$, $K_D = 19.25 \pm 2.40\ \mu M$), suggesting co-recognition of CD1d and the presented lipid antigen. Lipid-modulated recognition of CD1d by γδ TCRs has been observed by the 9C2 and DP10.7 Vδ1[+] TCRs, which directly contacted the

presented antigens via CDR3γ and CDR3δ loops, respectively[18,19]. Our experiments suggest γδ TCRs may recognise CD1d via two mechanisms, potentially binding the presented lipid or solely recognising the CD1d molecule itself.

### Vδ1[+] γδ TCRs recognise CD1d via similar binding modes

To identify whether the γδTCRs investigated here bound atop of CD1d or adopted more unusual ligand binding modes as seen in MR1 and CD1a reactive γδ TCRs, we performed in-line size-exclusion chromatography coupled small-angle X-ray scattering (SEC-SAXS) experiments (Fig. 1B, Supplementary Fig. 2)[21–23]. The γδ TCRs 7 and 8 co-complex samples were excluded from further analyses as the scattering profiles indicated a mixture of TCR and CD1d components and TCR dimers, rather than stable complexes in solution. The scattering profiles for the Vγ2 γδTCR3 and Vγ9 γδTCR6 were consistent with a globular protein. Compared with γδ TCRs 3 and 6, the γδ TCRs 2 and 8 scattering profiles were indicative of larger protein samples that suggested TCR dimerisation, with an oligomeric status of 1.8 and 1.5, respectively (Supplementary Table 2, Supplementary Fig. 2). This appears to be a conserved feature with Vγ5 TCRs such as γδTCRs 2 and 8, corroborating previous SAXS, X-ray crystallography and cryo-EM experiments[19,23,26,27]. Evaluation of the scattering profiles by comparison to known macromolecular structures of a Vγ5 TCR and Vγ5 TCR dimer confirmed that the γδTCRs 2 and 8 formed a dimer in solution (Supplementary Fig. 3). SAXS analyses of CD1d revealed scattering consistent with a globular monomer in solution.

Upon complexation with CD1d, γδ TCRs 2, 3 and 6 shifted on SEC, indicative of complex formation (Supplementary Fig. 3C). We next generated, aligned and averaged multiple ab initio reconstructions of the γδTCR 2, 3 and 6-CD1d-'endo' complex samples. The ab initio reconstructions revealed oblate particles that closely replicated the 9C2 γδ TCR-CD1d complex, suggesting the endo-reactive γδ TCRs 2, 3 and 6 recognised CD1d similarly to 9C2 (Fig. 1B)[19]. To establish the γδTCR interaction mode for CD1d, we compared the measured scattering profiles with theoretical scattering profiles derived from known macromolecular structures of γδTCR-MHC-like complexes, including the 9C2 γδ TCR in complex with CD1d-α-GalCer structure[28–30]. These analyses strongly supported an on top-docking geometry (Chi values) for γδ TCRs, 2, 3 and 6, as the scattering profiles closely matched the theoretical curve for 9C2 and a poor alignment of the G7 γδTCR-MR1 complex, where the TCR docked underneath the antigen binding platform (Supplementary Fig. 3)[23]. Collectively, our SPR and SEC-SAXS experiments revealed dual mechanisms for γδ TCR recognition of CD1d, including lipid antigen-dependent or -independent means that adopt similar 'end-to-end' docking modes.

### CD1d-mediated Vδ1[+] γδ T cell activation

To assess whether lipid antigen-dependent or -independent mechanisms for CD1d reactivity contributed to differing cellular activation outcomes, we transduced the Vδ1[+] γδ TCRs into a Jurkat.76 (J76) reporter cell line and assessed T cell activation by CD69 upregulation and CD3 downregulation. We compared γδ TCR signalling to the 9C1 αβ TCR, as it adopts a similar orthogonal binding mode atop CD1d as other previously known CD1d-restricted γδ TCRs[31]. In the absence of lipid pulsing, all Vδ1[+] γδ TCRs upregulated CD69 in response to CD1d expressing K-562 lymphoblast cells although to differing extents but did not respond to wildtype K-562 cells (Fig. 2A, Supplementary Fig. 9A). The γδ TCRs 2, 3, 7 and 8 expressing J76 cells upregulated CD69 following CD1d stimulation without a specific lipid antigen, whereas γδ TCR 6 although statistically significant did not appreciably upregulate CD69 compared to the other γδ TCRs, which mirrored negligible CD3 downregulation following CD1d stimulation. Whereas CD3 down-regulation was observed for the γδTCRs 2, 3, 7 and 8 expressing J76 cells, suggesting immune synapse formation following CD1d stimulation. We next probed whether α-GalCer treatment

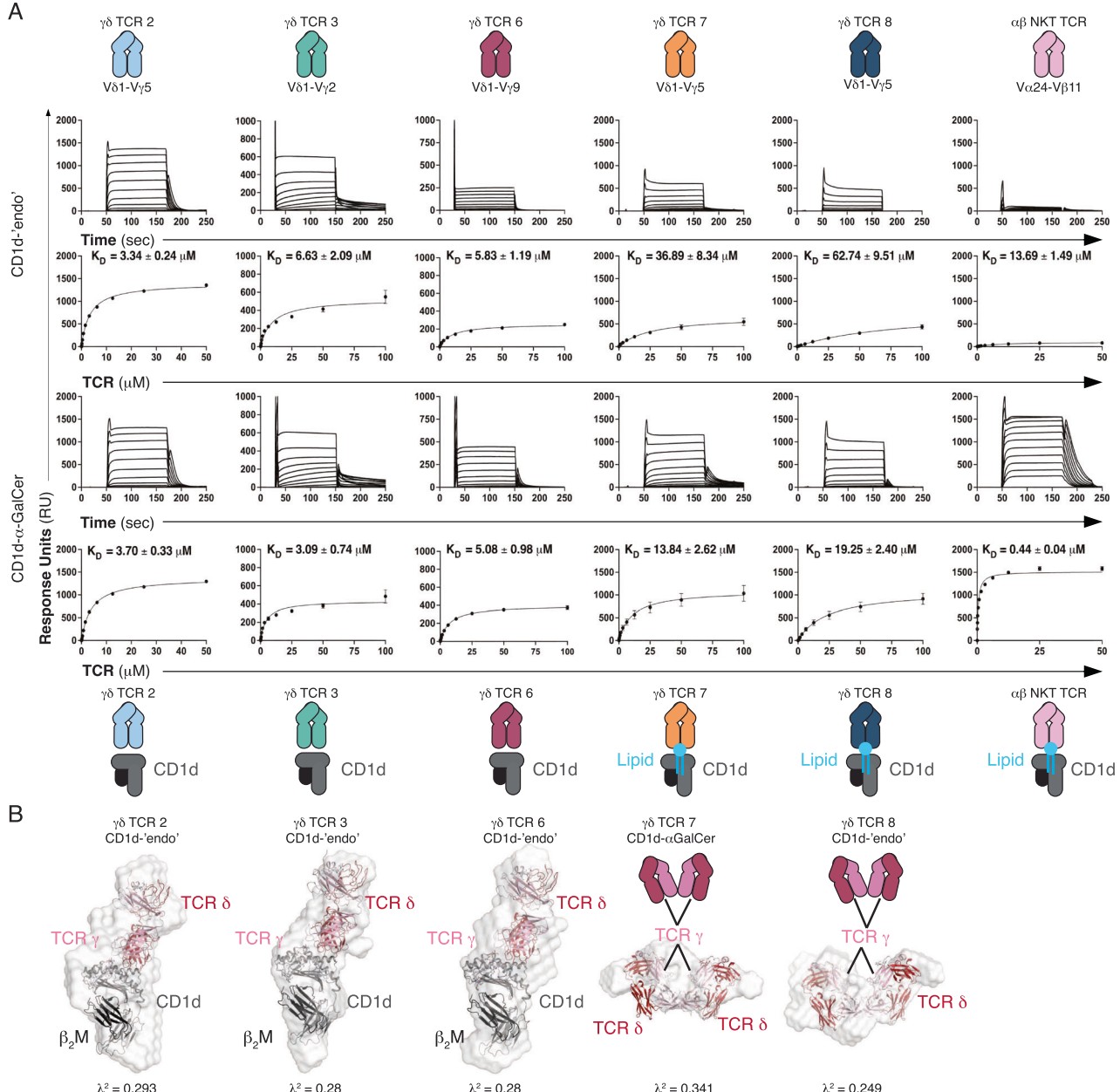

**Fig. 1 | CD1d recognition by Vδ1 + γδ TCRs. A** Affinity measurement analysis of γδ TCRs 2, 3 and 6, 7, 8 and the αβ NKT TCR determined by SPR. Sensograms are representative of a single dilution series. Error bars denote S.E.M., $K_D$ was determined from two independent experiments, performed in duplicate, using a 1:1 binding model. **B** SEC-SAXS analysis of γδ TCRs co-complexed with CD1d with ab initio models aligned with the 9C2 γδ TCR-CD1d complex or individual TCR and CD1d components. The 9C2 γ/δ chains were coloured in light and dark red, respectively, CD1d dark grey, β2M in black, and the ab initio reconstruction shown in white.

impacted Vδ1+ γδ T cell activation, which yielded no major alteration in activation (Fig. 2B). In comparison, the lipid-specific 9C1 αβ TCR upregulated CD69 at even the lowest concentration of α-GalCer, with a dose-dependent increase (Fig. 2B, Supplementary Fig. 9B). To further investigate the divergence between our steady-state affinity measurements and T cell stimulation capacity, we investigated the temporal impact of T cell signalling as noted to affect MR1-reactive γδ T cell activation[32]. We measured γδTCR-expressing J76 cell activation by detecting Nur77 upregulation after 2 h of stimulation with CD1d-expressing K-562 cells. Here, γδ TCR 2 and 3 had the highest frequency of Nur77+ cells, followed by γδ TCRs 7 and 8, which were comparable to the well-characterised 9C2 γδ TCR (Fig. 2C, Supplementary Fig. 10). γδ TCR 6, which had the lowest level of CD69 upregulation, also failed to upregulate Nur77 despite specific TCR reactivity to CD1d as attested by SPR. This may stem from some γδTCRs requiring high antigen levels to

trigger cellular activation[32]. Further, TCR-ligand affinity may influence γδ TCR signalling outcomes, as the highest affinity γδ TCRs 2 and 3 had the highest frequency of Nur77+ cells and CD3 downregulation. Collectively, these experiments indicated that the Vδ1+ TCR panel activated in response to CD1d-expressing cells independently of a specific lipid antigen, suggesting a disconnect between steady-state affinity and T cell activation capacity.

## CD1d-induced γδ TCR clustering

To identify whether differences in T cell activation were associated with γδTCR-CD3 triggering and proximal signalling events, we performed single-molecule imaging with CD1d-endo and CD1d-α-GalCer using a supported lipid bilayer system (Fig. 3A, B). In response to CD1d-endo, of the γδ TCRs we investigated, only γδ TCR 6 failed to undergo significant changes to TCR cluster number, cluster area or percentage

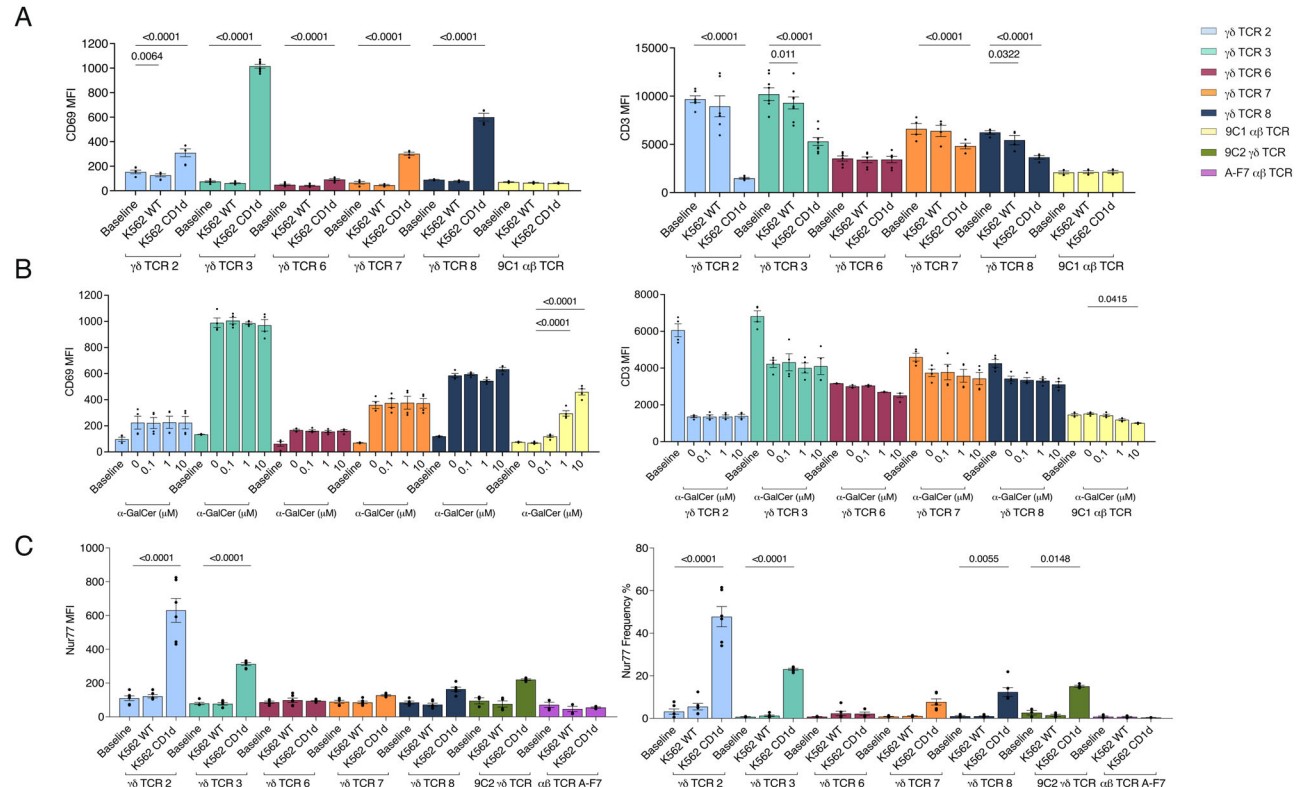

**Fig. 2 | CD1d activates γδ T cells. A** Mean Fluorescence Index (MFI) of γδ TCR transduced Jurkat.76 cells after 16 co-cultures with K-562.WT and K-562.CD1d cells assessing CD69 upregulation and CD3 downregulation. Significance is comparing Baseline, Jurkat cells alone, to the co-culture. Data was generated from two independent experiments, performed in duplicate. **B** CD69 and CD3 MFI following serial dilution of α-GalCer with K-562.CD1d cells, assessing CD69 and CD3 levels compared to vehicle, 0 μM of α-GalCer with K-562.CD1d cells. Data was generated from two independent experiments, performed in duplicate. **C** Jurkat T cell Nur77 MFI and Frequency were assessed following 2 h of co-culture with either K-562.WT or K-562.CD1d cells. Data was generated from three independent experiments, performed in duplicate. Statistical analysis was performed with one-way ANOVA with Tukey's multiple comparison test. Error bars denote S.E.M.

of TCR localisations compared to unstimulated ICAM-1 alone (Fig. 3C, Supplementary Fig. 4). This mirrored our CD69 and Nur77 experiments, where γδ TCR 6 was weakly reactive to CD1d. Indeed, following activation, the γδTCRs 2, 3, 7 and 8 underwent the greatest extent of TCR clustering across both concentrations of CD1d-endo tested. Notably, all γδ TCR cell lines tested showed increased TCR cluster area and enhanced TCR triggering, with increased CD1d ligand abundance, a feature also noted with a MR1-reactive γδ TCR[32]. As SPR indicated that γδ TCR7 and 8 had a higher affinity for CD1d-α-GalCer compared to CD1d-endo, we tested CD1d-α-GalCer effects on γδ TCR clustering and proximal signalling (Fig. 3B, D). As a control, we included the 9C1 αβ TCR, which underwent TCR clustering upon engagement with CD1d-α-GalCer, but not CD1d-endo or ICAM-1 (Fig. 3C, D). Again, we observed increased TCR clustering with increasing amounts of CD1d-α-GalCer (Fig. 3D, Supplementary Fig. 4B). Across all cell lines, apart from γδ TCR 7, there was an increase in activated TCR clustering with CD1d-α-GalCer compared to CD1d-endo. Collectively, our single-molecule imaging experiments show that CD1d-endo is sufficient to induce γδ TCR clustering and phosphorylation, although TCR triggering was enhanced with higher-affinity lipid antigens and increasing antigen concentrations. The remarkable flexibility of the γδ TCR-CD3 signalling apparatus may require higher CD1d ligand densities to stabilise the complex and promote TCR signalling across shorter time points, as seen with MR1 reactive γδ TCRs[32].

### Impact of Cγ Exon insertions on γδ T cell activation

Further complicating our understanding of γδ TCR triggering are allelic variants of the γδ TCR constant domain. The TCR γ locus encodes two constant region genes, a *Trgc1* (Cγ1) which consists of

three exons, and a *Trgc2* gene that contains 4, or rarely five exons stemming from exon 2 duplications and triplications, respectively[33]. The duplication and triplication events of exon 2 result in a 16 and 32-amino acid extension of the γδ TCR connecting peptide and a C > W mutation that prevents disulphide formation with Cδ. Extended connecting peptides were recently shown to endow the γδTCR with extreme dynamism within the CD3 signalling apparatus, which in turn modulated proximal signalling[26,32]. Xin et al. recently demonstrated that in the exon 2 duplication event Cγ2, dampened γδ TCR activation compared to the Cγ1[26]. We investigated whether the exon 2 triplication event, Cγ3, had a further impact on γδ TCR signalling outcomes, using the previously well-characterised CD1d-restricted 9C2 γδ TCR[19,26]. Following stimulation by K-562.CD1d cells, the Cγ1 allele led to the greatest CD69 upregulation, with a marked reduction in CD69 production by the Cγ2 allele (Fig. 4A, Supplementary Fig. 11A). The Cγ3 allele completely inhibited γδ T cell activation, as it did not upregulate CD69 above baseline levels. As the 9C2 γδ TCR has improved affinity for CD1d presenting α-GalCer ($K_D = 16$ μM) compared to CD1d-'endo' ($K_D = 35$ μM), we reasoned that the antigen may serve to stabilise TCR-CD1d interactions leading to improved signalling outcomes[19]. While α-GalCer increased CD69 upregulation with 9C2 Cγ1, it did not improve signalling with either Cγ2 or Cγ3 (Fig. 4B, Supplementary Fig. 11B). Therefore, elongation of the γδ TCR-CD3 complex via Cγ Exon 2 duplication and triplication events reduced and even ablated T cell activation. This raised questions on whether the Cγ2 and Cγ3 alleles could form functional γδ TCR signalling complexes.

To address this, we performed single-molecule imaging on the 9C2 Cγ alleles (Fig. 4C). Although Cγ2 and Cγ3 displayed reduced capacity to upregulate CD69 compared to Cγ1, both readily formed

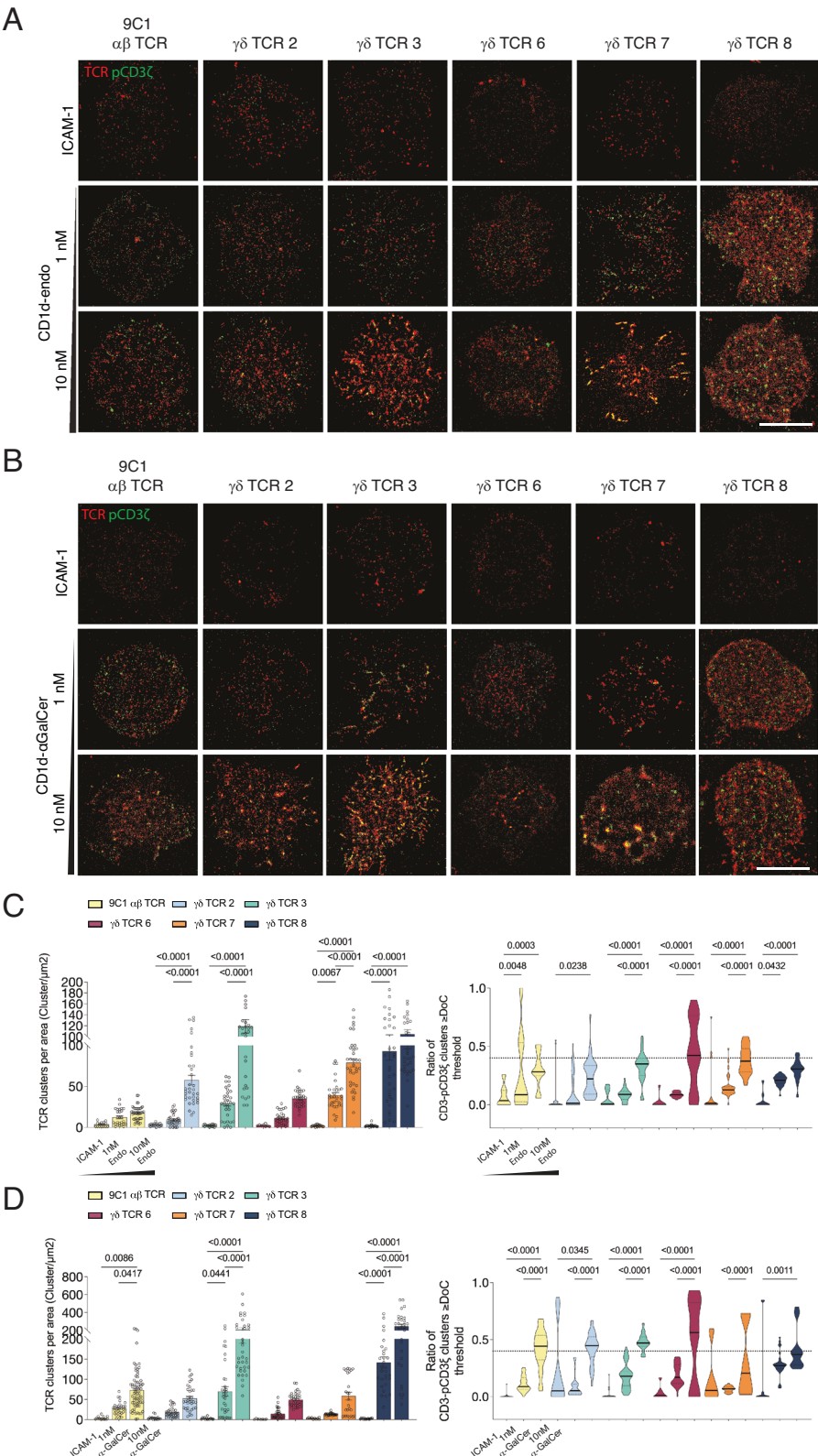

**Fig. 3 | CD1d-induced TCR clustering and CD3 phosphorylation.** dSTORM images of CD1d-reactive αβ TCR 9C1 and CD1d-reactive γδ TCRs expressed in Jurkat T cells, stimulated with ICAM-1 alone, ICAM-1 + CD1d-endo (**A**) or ICAM-1 + CD1d-α-GalCer (**B**) on supported lipid bilayers (SLBs) at varying concentrations. Cells were stained for CD3 (AF647, red) and CD3ζ/pCD3ζ (AF568, green). ICAM-1-only SLBs served as antigen-free unstimulated controls. Scale bar: 5 μm. **C**, **D** Cluster analysis of dSTORM images using DBSCAN and Clus-DoC. DBSCAN and Clus-DoC Analysis were performed across $n \geq 30$ single Jurkat T cells with three independent replicates. Statistical analysis was performed using one-way ANOVA with Tukey's multiple comparisons test against ICAM-1 controls. Error bars represent S.E.M.

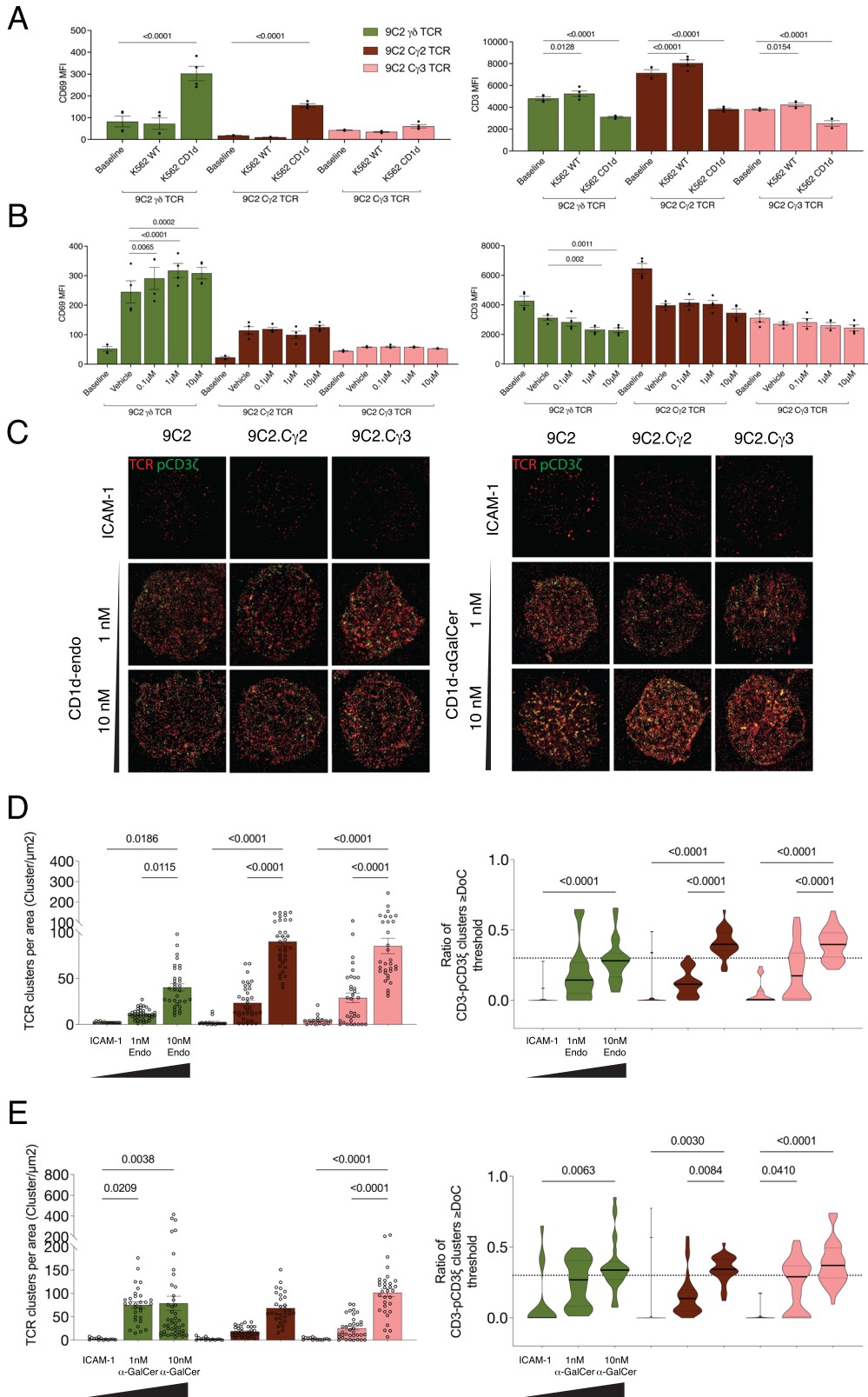

**Fig. 4 | Impact of Cγ Exon 2 insertions upon γδ T cell activation. A** MFI of CD69 and CD3 of 9C2 γδ TCR Cγ alleles transduced into Jurkat cells. Activation was measured following co-culture with either K-562.WT or K-562.CD1d cells after 16 h of stimulation. **B** Assessment of Jurkat T cells CD69 and CD3 surface expression post 16-h stimulation with K-562.CD1d cells treated with dilutions of α-GalCer. **C** dSTORM images of 9C2 Cγ alleles stimulated with ICAM-1, ICAM-1 + CD1d-endo or ICAM-1 + CD1d-αGalCer. Cells were stained with α-CD3 (red) and α-CD3ζ/p-CD3ζ (green). Cluster analysis of CD1d-endo (**D**) and CD1d-αGalCer (**E**) stimulated cells. Analysis was performed as outlined in Fig. 3, using one-way ANOVA with Tukey's multiple comparison test. Analysis was performed on two independent experiments, with duplicate wells. Error bars denote S.E.M.

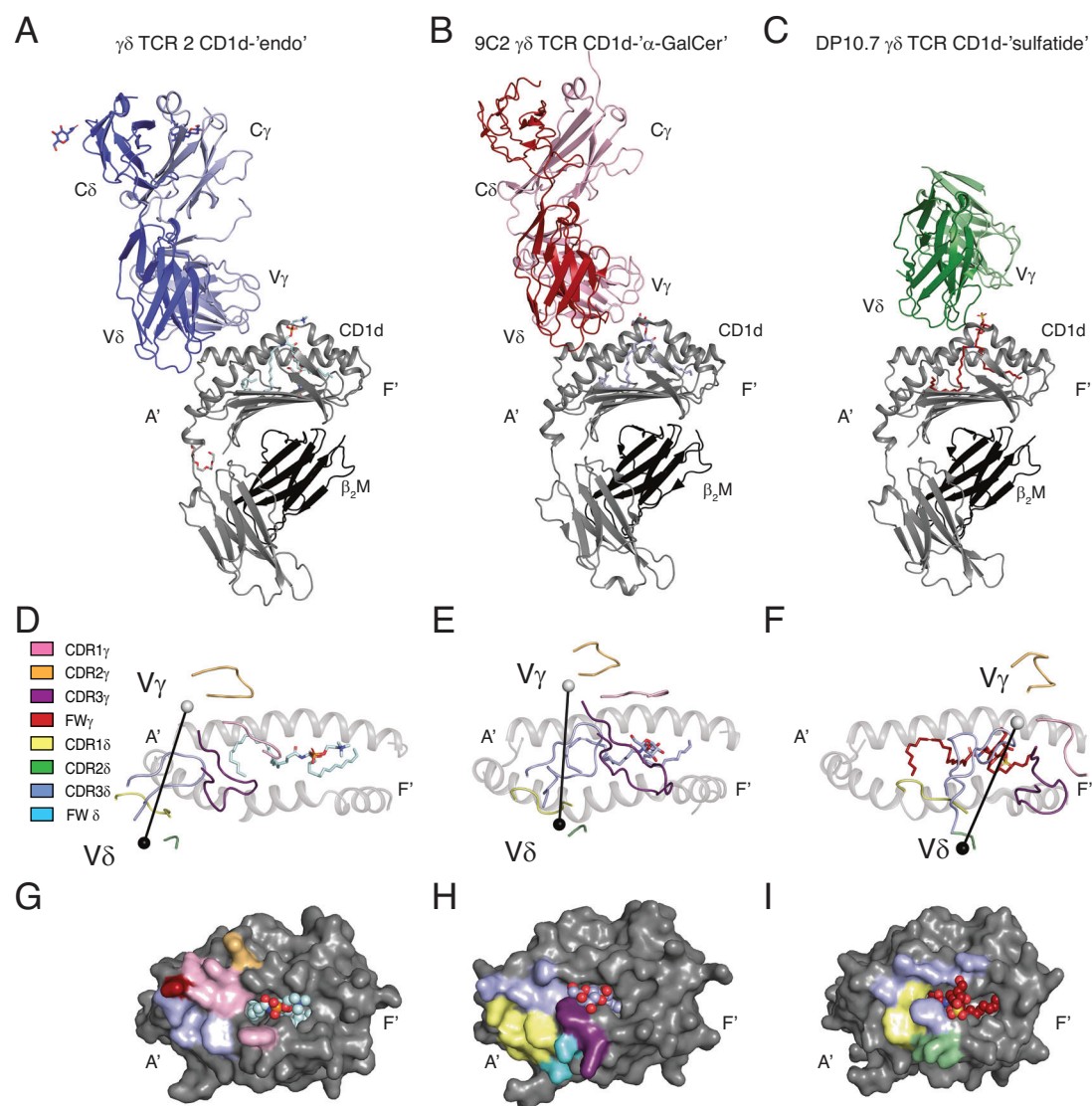

**Fig. 5 | Structural overview of the γδ TCR-CD1d complex.** Cartoon representation of γδ TCR 2 bound to CD1d-'endo'(**A**), 9C2 γδ TCR bound to CD1d-'α-GalCer'(**B**) (PDB ID: 4LHU) and DP10.7 γδ TCR bound to CD1d-'sulfatide' (**C**) (PDB ID: 4MNG). TCR γ/δ chains are coloured light/dark variants of blue, red and green. CD1d is shown in grey, β₂M in black. CDR loop positions and V chain COM shown for γδ TCR 2 (**D**), 9C2 (**E**) and DP10.7 (**F**). Vδ and Vγ COM are coloured in black and white, respectively. CDR loops are coloured pink-CDR1γ, orange-CDR2γ, purple-CDR3γ, red-FWγ, yellow-CDR1δ, green-CDR2δ, blue-CDR3δ and light-blue-FWδ. FW denotes framework contacts. Molecular contacts on CD1d shown for γδ TCR 2 (**G**), 9C2 (**H**) and DP10.7 (**I**), coloured as seen in (**D–F**). Lipid antigens are either depicted as sticks (**A–F**) or spheres (**G–I**).

functional, similarly sized, TCR clusters against CD1d-endo and CD1d-α-GalCer indicating the Cγ exon duplication and triplications do not prevent TCR triggering (Fig. 4C–E, Supplementary Fig. 5). Instead, the exon 2 insertions may increase the length and flexibility of the membrane proximal connecting peptides that link the γδ TCR extracellular domains to the CD3 signalling apparatus. This may limit efficient TCR signal transduction and ultimately cellular activation, corroborating evidence that γδ TCR flexibility and connecting peptide length impact γδ T cell activation[26,32]. Additionally, γδ TCR docking modalities are thought to be critical for functional γδ TCR signalling[23]. Yet, how γδ TCRs engage their ligands remains largely unclear and can vary dramatically across the same ligand[22,23].

### Overview of the γδ TCR 2 CD1d-'endo' complex
To gain insight into the molecular basis of γδ TCR CD1d lipid-independence, we determined the X-ray crystal structure of γδ TCR 2 bound to CD1d-'endo'(Supplementary Table 3). Although we determined the structure of the γδ TCR-CD1d complex with CD1d carrying a mixture of endogenous lipids from the mammalian expression system, density was observed within the antigen-presenting cleft of CD1d, indicating the abundance of one class of lipid antigen. Here, we modelled sphingomyelin, which has previously been identified as a component of the CD1d-'endo' lipid repertoire (Supplementary Fig. 6A, B)[34,35]. Despite sphingomyelin being modelled into the crystal structure, we cannot exclude the possibility that our crystal structure likely contains a mixture of phospholipid antigens.

The γδ TCR 2 docked over the extreme A' end of CD1d antigen-binding cleft (Fig. 5A). This docking geometry was similar to the 9C2 γδ TCR, which shares the same Vδ1 Vγ5 TCR chain usage, that bound over the A' of CD1d (Fig. 5B, E, H), and distinct from the Vδ1Vγ4 DP10.7 γδ TCR, which bound centrally over the protruding sulfatide lipid head group (Fig. 5C, F, I). The γδ TCR 2 bound orthogonally atop CD1d (~80°), with the Vγ and Vδ chains positioned centrally over the α1 and α2 helices, respectively (Fig. 5D, G).

This more 'conventional' γδ TCR 2 recognition mode differed from the perpendicular, side, and underneath docking observed for

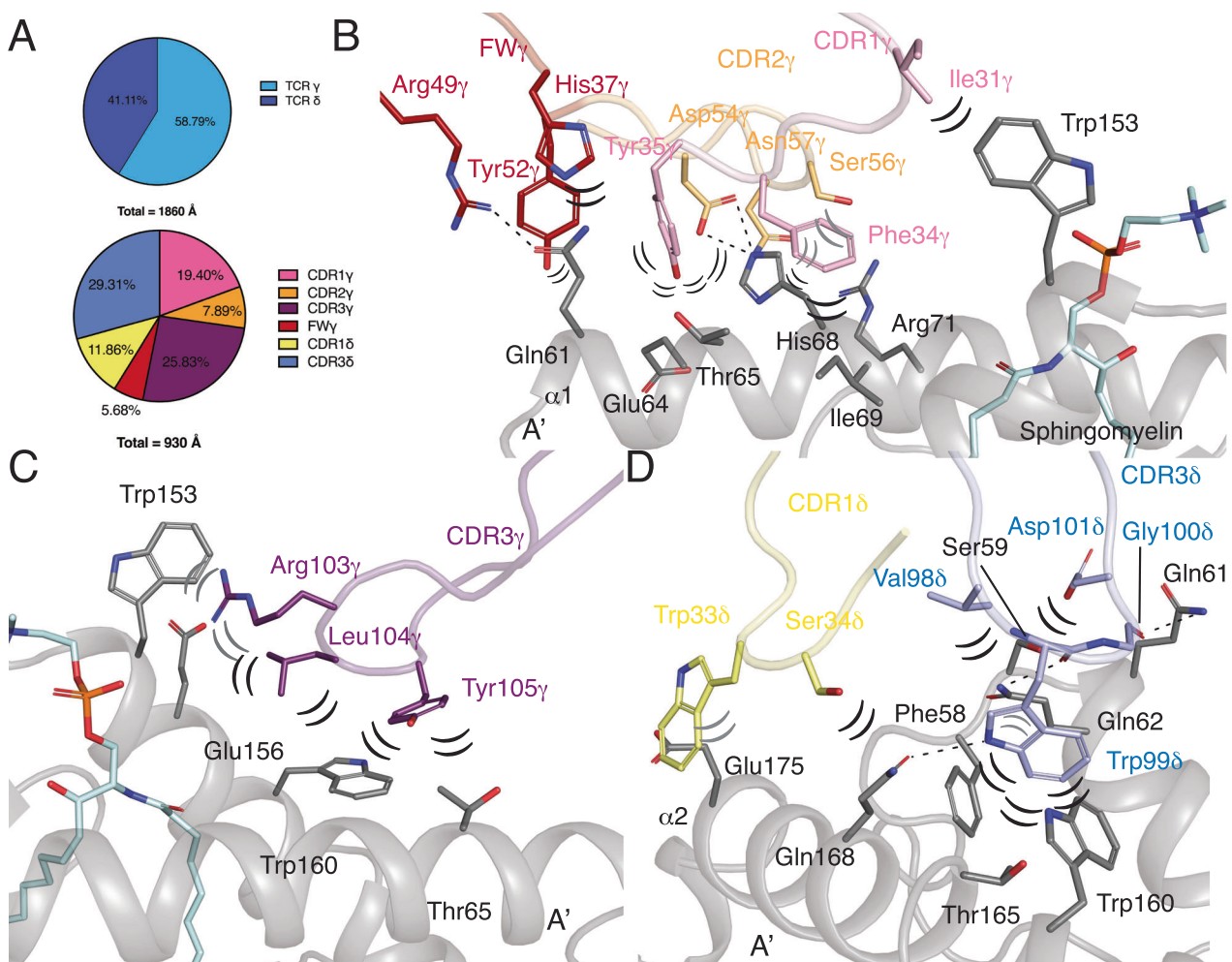

**Fig. 6 | Molecular interface of the γδ TCR 2 CD1d interaction. A** Buried Surface Area and CDR loop contributions of the γδ TCR 2-CD1d interface. Interactions between γδ TCR 2 and CD1d via FWγ, CDR1γ and CDR2γ (**B**), CDR3γ (**C**) and CDR1δ, CDR3δ (**D**). CDR loops are coloured pink-CDR1γ, orange-CDR2γ, purple-CDR3γ, red-FWγ, yellow-CDR1δ, green-CDR2δ and blue-CDR3δ. FW denotes framework contacts. CD1d is shown in dark grey. Sphingomyelin lipid is shown as sticks. Dotted lines denote hydrogen bonds, parentheses indicate Van der Waals interactions.

the recently resolved MR1 and CD1a γδ TCR complex structures[21–23]. However, the extreme A' binding mode was unusual, with the Vδ/Vγ shifting 10 Å/6 Å with respect to the 9C2 TCR and 23 Å/26 Å compared to the DP10.7 γδ TCR (Supplementary Fig. 6C). Thus, the CDR loop positions and molecular contacts of the γδ TCR 2 to CD1d are unique to this complex, and different from other γδTCR and αβTCR-CD1d complexes. The CD1d-γδ TCR 2 complex further demonstrates the diversity of γδ TCR antigen recognition even across γδ TCRs with the same gene usage.

### γδ TCR 2 binds CD1d independently of lipid antigens
The total buried surface area (BSA) of the TCR-CD1d interface was ≈ 1860Å (Fig. 6A). The TCR chains contributed asymmetrically to the interface, with ≈ 59% of the interface mediated by TCR-γ and ≈ 41% by TCR-δ (Fig. 6A). This contrasts with the emerging trend of Vδ2⁻ γδ TCR-ligand recognition being dominated by the TCR-δ chain, in particular CDR3δ[36,37].

Notably, FWγ contributed to this interface with several polar interactions to CD1d (Fig. 6B, Supplementary Table 4). This served to stabilise CDR1γ and CDR2γ, which were positioned on either side of the CD1d-α1 helices and made a number of key contacts to CD1d (Fig. 6B).

A key determinant in lipid binding for the 9C2 Vδ1 Vγ5 TCR was CDR3γ, namely Arg103γ and Tyr111γ, that directly contact the head-group of α-GalCer. Instead, the CDR3γ of γδ TCR 2 did not contact the

lipid antigen (Fig. 6C), providing the molecular basis for the lipid-independent binding identified in the SPR experiments. Rather, CDR3γ solely contacted the CD1d molecule itself (Fig. 6C). The 9C2 TCR and γδTCR 2 possess different CDR3γ loop lengths, 18 versus 13 amino acids, which may account for the differences identified in biochemical and structure-based observations. The shorter CDR3γ loop of γδ TCR 2 leads to a rearrangement of the TCR-CD1d interface compared to the 9C2 TCR, shifting the CDR3γ/δ loops ~ 8 Å towards the A' of CD1d, preventing direction recognition of the presented lipid antigen.

TCR-δ chain binding was predominantly mediated by the CDR3δ loop. CDR2δ did not contact CD1d while CDR1δ made minimal contacts, a feature not seen in other CD1d-γδ TCR complexes that relied heavily on CDR1δ for CD1d engagement[18,19]. Central to this recognition was Trp99δ, which wedged in between the α helices of CD1d, forming an extensive network of interactions with CD1d (Fig. 6D). Similar to CDR3γ, the CDR3δ loop of γδ TCR 2 is shorter compared to 9C2, 12 vs 16 amino acids. Consequently, neither CDR3γ nor CDR3δ made contact with the presented lipid antigen, providing structural evidence for lipid-independent CD1d binding by a γδ TCR.

### A conserved aromatic zone on CD1d
γδ TCR 2 docking atop CD1d was reminiscent of a single domain antibody (VHH) 1D12, which also bound over the A' distal end of CD1d and make no contacts to the presented lipid antigen[38]. VHH1D12 bound

with extremely high nanomolar affinity and served to stabilise NKT TCR binding to CD1d, promoting cellular activation. Akin to Trp99δ of the γδ TCR 2, Phe29 from VHH1D12-CDR1 served as a lynchpin of binding, interacting with Phe58, Gln62, Trp160, Thr165 and Gln168, which were all binding partners with Trp99δ (Supplementary Fig. 7A, B). We then analysed other γδ and atypical αβ NKT CD1d-TCR complexes that similarly bound over the A' end of CD1d to determine whether this aromatic zone was a conserved binding motif. Indeed, CDR1δ of both the γδ 9C2 and 9B4 δ/αβ TCRs bound the same motif as CDR3δ of γδ TCR 2, suggesting recognition of this motif serves as a key determinant of TCR-ligation (Supplementary Fig. 7C, D). Although the CDR1δ of DP10.7 TCR was shifted more centrally, it was pincered by Trp153 and Trp160, providing further evidence aromatic interactions are central to γδ TCR recognition of CD1d (Supplementary Fig. 7E). The atypical αβ NKT TCRs, 9C1 and 9B2 similarly interacted with the A' of CD1d, with Phe111 of CDR3α and CDR3β of 9C1 and 9B2, respectively, interacting with Trp160 of CD1d (Supplementary Fig. 7F, G). This suggests that the A' of CD1d, containing a large non-polar patch, serves as a key TCR binding site for atypical αβ NKT and γδ TCRs that is primarily governed by CDR loops containing aromatic residues (Supplementary Fig. 7H)[31].

## Discussion

Here, we provide further evidence that γδ TCRs adopt diverse antigen recognition strategies, either recognising the CD1d molecule itself or binding both CD1d and the lipid antigen[37]. The Vδ1-Vγ5 TCRs investigated here displayed two modes of interaction being CD1d-endo reactive (γδ TCR 2) or displayed a lipid-modulated interaction (γδ TCR 7 and 8), indicating that γδ TCR recognition strategies are potentially clonally dependent. Although γδ TCR clonal expansion has been identified, whether this occurs in an MHC-I-like dependent manner remains unclear[7,10].

The recent elucidation of the γδ TCR-CD3 complex via cryo-EM highlighted the flexibility of the γδ TCR ectodomain within the signalling apparatus, which was resolved as a consequence of Vγ5 chain dimerisation, which stabilised the TCR ectodomain[26,32]. In our SEC-SAXS experiments, we noted dimerisation of two Vδ1-Vγ5 TCRs, a feature that was absent in Vγ9 and Vγ2 Vδ1+ TCRs. Mutagenesis of the dimerisation interface in the 9C2 γδ TCR ablates antigen-driven CD69+ upregulation despite the TCR-CD3 complex maintaining its functionality[26]. One potential explanation is that Vγ5+ TCRs dimerise to promote TCR-mediated signalling in the absence of high-affinity ligand interactions. Hypo-reactive γδ TCRs are a recurring theme, with γδ TCRs reactive towards MR1 and CD1a being poor activators despite binding their ligands with moderate affinity[21,23]. Whether such phenomena occur in primary γδ T cells is unclear, but raises further questions on the mechanisms underpinning γδ TCR triggering, particularly if other Vγ pairings induce dimerisation to enhance TCR signal propagation.

Similar to MR1-, CD1b- and CD1a-reactive γδ T cells, CD1d-restricted γδ T cells activated in the absence of lipid antigen stimulation, suggesting CD1d ligand availability is a regulator of γδ T cell activation[21,23,25,32]. A limitation of this study is the use of Jurkat T cells to investigate γδ T cell activation, raising questions on whether our conclusions extend to primary γδ T cells. We have previously demonstrated that PBMC-derived CD1d-restricted γδ T cells activate in response to cells overexpressing CD1d in the absence of lipid pulsing[19]. This alludes to CD1d-restricted primary γδ T cell activation also being dependent on ligand availability, although further experiments are required to confirm these preliminary findings.

Ligand availability limiting γδ T cell activation is potentially resultant of the increased flexibility of the γδ TCR within the γδ TCR-CD3 signalling complex, therefore increasing the number of ligand interactions required to stabilise the γδ TCR-CD3 complex to promote signal transduction[26,32]. The Cγ alleles further add to this flexibility, but do not impede TCR ligand binding[26]. However, increasing the length of the γδ TCR-CD3 complex via the Cγ2 and Cγ3 alleles reduced and impeded cellular activation, respectively, despite maintaining their ability to form competent γδ TCR-CD3 complexes. This further highlights how, in isolation, γδ TCR ligand affinity is a poor predictor for γδ T cell functional outcomes, as γδ TCR flexibility and ligand binding modes have also been shown to regulate γδ T cell activation[21,23,26,32].

We next determined the structure of a Vδ1-Vγ5 TCR bound to CD1d-'endo'. The γδ TCR2 was situated over the A' roof of CD1d that resulted in the absence of γδ TCR-lipid interactions. This extends an emerging theme of γδ TCRs displaying antibody-like ligand recognition[21–23].

The A' lipid-independent docking by γδ TCR 2 was similar to the BK6 and 3C8 αβ TCRs, which bound CD1a and CD1c, respectively[39,40]. Whilst both BK6 and 3C8 bound the A' roof of CD1a/c, neither αβ TCR contacted the presented lipid antigens, diverging from the CD1-lipid co-recognition paradigm[39,40]. In the absence of direct lipid interactions, presentation of 'non-permissive' lipids that disrupt TCR-CD1 binding serves to regulate αβ T cell activation[41]. Divergence from the co-recognition model has recently been observed in type I NKT cells, which recognised small headless ceramide lipids presented by CD1d[42]. Primarily recognising CD1d, the limited contacts of the NKT TCR to the presented ceramide coincided with reduced cellular staining and activation compared to the prototypic α-GalCer antigen. Deciphering whether γδ T cells will recognise antigens in a similar lipid-reactive or co-recognition-dependent manner is unpredictable. For instance, Vδ1+ TCR complexes display remarkable diversity in binding modalities. The Vδ1+ γδ TCR 2-CD1d complex determined here continues the trend of breaking the co-recognition paradigm as seen in MR1 and CD1a TCR-ligand complexes, and differs from the 9C2 and DP10.7 γδ TCRs that co-recognise the presented lipid antigen and CD1d molecule[18,19,21–23].

Our findings illustrate that γδ TCRs can recognise CD1d via diverse mechanisms and bind irrespective of the lipid antigen. This represents the structural characterisation of a γδ TCR bound to CD1d in a lipid-independent manner. This further illuminates the complexity in understanding γδ T cell activation and their roles within the immune response more broadly.

## Methods
### Protein production and purification

The γδ TCRs and CD1d constructs were designed and expressed in Human Embryonic Kidney (HEK) 293 F cells (Gibco) and purified as previously described[19]. Both γδ TCR and CD1d constructs contained C-terminal His-Tags, Fos-Jun Zippers and a BirA sequence. In brief, HEK-293F cells were transfected with either CD1d-$\beta_2$M or TCR-γ and TCR-δ chains and allowed to express for 5 days at 37 °C, 5% $CO_2$. On days 1 and 3, media was supplemented with 1 mM NEAA, 1 mM Gluta-MAX and 33 mM Glucose. On day 5, the transfected culture was harvested, centrifuged at 4000 g at 4 °C, and the supernatant dialysed against 10 mM Tris-HCl, pH 8.5, 300 mM NaCl. The dialysed samples were then purified via Ni-NTA, followed by size exclusion chromatography to yield homogeneous pure protein. Prior to biochemical or crystallography-based experiments, the C-terminal His-tag and Fos-Jun zippers were removed by thrombin digestion, and the protein was further resolved by size exclusion chromatography. For surface plasmon resonance experiments, CD1d-$\beta_2$M was expressed in High-Five insect cells (maintained in house > 10 years), purified and biotinylated as previously described[43]. The αβ NKT.15 TCR was expressed in bacterial cells, refolded and purified as previously described[43].

### Crystallisation and structural analysis

Prior to crystallisation, HEK-293F Gnti-/- produced γδ TCR 2 and CD1d-'endo' were incubated overnight at 4 °C at a 1:1 molar ratio in the presence of EndoH (New England Biolabs). To resolve

co-complexes from individual components, SEC was performed via a SE 200 10/30 (GE Healthcare). Co-complexed fractions were pooled and concentrated to 8 mg/mL for crystallisation experiments at the Monash Molecular Crystallisation Platform. Crystals of the γδ TCR 2-CD1d-'endo' complex formed in 16% PEG 3350, 0.05 M CBTP, pH 5.0. Individual crystals were cryoprotected in mother liquor with the addition of 40% PEG 400 and flash frozen. Data was collected on the MX2 beamline at the Australian Synchrotron[44]. Data were processed using XDS and CCP programme suites. Molecular replacement was performed using Phenix[45], with the individual components of the γδ TCR, with the CDR loops removed in COOT (PDB ID: 4LHF) and CD1d-β2 M, with the lipid antigen removed (PDB ID: 8SOS), were used as individual search models[46]. A single γδ TCR-CD1d complex was present in the asymmetric unit. Manual model building was performed in COOT and further refined in Phenix. Buried surface area was calculated by PDBePISA, and molecular interactions were analysed via CONTACT from the CPP4 Software Suite[47]. The γδ TCR 2-CD1d complex was refined and deposited in the Protein Data Bank for structural validation.

### Surface plasmon resonance

Surface plasmon resonance (SPR) experiments were performed on a BIACore T3000 (γδ TCRs – 2, 7 and 8) and T200 (γδ TCRs – 3 and 6) in TBS Buffer (10 mM Tris-HCl, pH 8.0, 150 mM NaCl) with 0.5% bovine serum albumin (BSA) at 25 °C. The αβ NKT TCR, NKT.15, was used as a positive control for all SPR experiments. Biotinylated CD1d and CD1b proteins were coupled to a streptavidin chip (GE Healthcare) to approximately 1000–2000 response units (RU). Mammalian expressed γδ and refolded αβ NKT TCRs were used as the analyte and serially diluted from 200 to 0 μM. Sensograms and TCR affinity curves were generated and analysed in GraphPad Prism 10. Experiments were performed twice with duplicate injections.

### Size-Exclusion Chromatography small-angle X-ray scattering (SEC-SAXS) data collection

SEC-SAXS experiments were performed on the SAXS/WAXS beamline at the Australian Synchrotron with co-flow to minimise radiation damage and an in-line SEC[48]. Prior to the SEC-SAXS experiment, γδ TCRs and CD1d were incubated overnight at 4 °C at a 1:1 molar ratio. Approximately 60 μL of each sample, ranging from 5–10 mg/mL, were injected onto a Superose 6 5/150 increase column (GE Healthcare), in 10 mM Tris-HCl, pH 8.0, 150 mM NaCl. SEC-SAXS images were analysed in BioXTAS RAW[30]. Prior to analysis, images were buffer-subtracted, and to aid ab initio model interpretation, only images containing co-complexed samples, as determined by Radius of Gyration (Rg) and Porod Volume (V_P), were kept for further analysis. In BioXTAS RAW, the raw scattering SEC-SAXS scattering curves, Guinier analysis, P(r) distribution plots, Rg and maximum particle dimension (Dmax) were determined. A summary of the SEC-SAXS data collection can be found in Supplementary Fig. S2, Supplementary Table S2. Ab initio models were generated utilising DAMMIF in BioXTAS RAW; 15 reconstructions were generated, averaged and refined via DAMAVER and DAMMIN. X-ray crystallography structures were aligned to the generated reconstruction. CRYSOL was then performed to compare the theoretical scattering from the aligned model to the raw scattering curves[49].

### Generation of stable TCR cell lines

γδ and αβ TCRs were cloned into pMIG-II (pMSCV-IRES-GFP II), with the individual TCR chains separated by a P2A linker. Parental Jurkat-76 cells (maintained in house > 10 years), which lack endogenous αβ TCR expression and consequently CD3 cell surface expression, were retrovirally transduced with either γδ or αβ TCR genes. Jurkat-76 cells were maintained in RPMI supplemented with 10% FCS, 15 mM HEPES, 1 mM Sodium Pyruvate, 1x Non-essential amino acids and 2 mM

GlutaMax at 37 °C at 5% $CO_2$ (all sourced from ThermoFisher). After 2 rounds of transduction, successful TCR transduction was assessed via flow cytometry by the presence of GFP+ and staining of CD3+, α-CD3-PE (BD Biosciences, 1:200, #555333), cells. Multiple rounds of cell sorting were performed at the Monash FlowCore Facility to attain a homogeneous GFP+ CD3+ cell population.

### T cell activation assays

T cell activation was assessed via the upregulation of CD69 and the downregulation of CD3 and Nur77 expression. For CD69 upregulation detection experiments, transduced Jurkat.76 cells were co-incubated with K-562 cells for 16 h at 37 °C. The co-culture was then harvested, washed in PBS and stained with Zombie Aqua Live/Dead Stain (BioLegend). Cells were then washed with FACS Buffer and stained with α-CD1d-R710 (BD Biosciences, 1:200, #567984), α-CD3-PE (BD Biosciences, 1:200, #555333) and α-CD69-APC (BD Biosciences, 1:200, #555533) antibodies. For Nur77 experiments, transduced Jurkat.76 cells were incubated with K-562 cells for 2 h at 37 °C and then harvested. Cells were then stained with Live/Dead stain and stained with α-CD1d-R710 (BD Biosciences, 1:200, #567984) and α-CD3-PE (BD Biosciences, 1:200, #555333). After washing with FACS buffer, cells were resuspended and incubated in Fixation Buffer in the dark at room temperature. Cells were then permeabilised and stained with α-Nur77-AF647 (BD Biosciences, 1:200, #566735). CD69 and Nur77 upregulation was then assessed at the Monash FlowCore Facility on a BD Fortessa, cells were gated on Lymphocytes, Single cells, Live cells, CD1d- and CD3/GFP+. In GraphPad Prism v10, statistical analysis was performed via one-way ANOVA. Exact P-values are shown for all statistically significant comparisons; non-significant comparisons are not shown. Error bars denote S.E.M.

### Preparation of supported lipid bilayer (SLB)

Glass coverslips (0.17 mm thickness) were cleaned with 1 M KOH, rinsed with Milli-Q water, and dried in a fume hood after ethanol treatment. Following plasma cleaning, coverslips were attached to eight-well silicon chambers (ibidi, #80841). SLBs were prepared using vesicle extrusion of a 1 mg/ml liposome solution. The liposome composition included 96.5% DOPC, 2% DGS-NTA(Ni), 1% Biotinyl-Cap-PE, and 0.5% PEG5000-PE (mol %; Avanti Polar Lipids). Extruded liposomes were added to chambers (1:5 ratio with Milli-Q water containing 10 mM $CaCl_2$) and incubated for 30 min at room temperature before gentle PBS rinsing. Fluorescence recovery after photobleaching (FRAP) was used to examine SLB lateral mobility. Excess $Ca^{2+}$ ions were removed with 0.5 mM EDTA, followed by Milli-Q water rinsing. NTA groups were recharged with 1 mM $NiCl_2$ solution for 15 min, followed by PBS washing.

### Stimulation and immunostaining of T cells on SLB

Biotinylated SLBs were coupled with streptavidin (100 μg/ml) followed by biotinylated CD1d-α-GalCer or CD1d-endo (1–10 nM). NTA-functionalized lipids were coupled with His-tagged ICAM-1 (200 ng/ml). SLBs were rinsed and pre-incubated with warm RPMI medium. Jurkat TCR transductants were stimulated on SLBs for 5 min at 37 °C, fixed with 4% paraformaldehyde, permeabilized with 0.1% Triton X-100, and blocked with 5% BSA. Cells were immunostained with specific antibodies, namely anti-CD3ε-Alexa Fluor 647 (BioLegend, #300416, Clone UCHT1) (1:300 dilution) and anti-pCD3ζ-Alexa Fluor 568 (BD Biosciences, #558402) (1:300 dilution) for 1 h at room temperature. After washing, a final fixation step was performed, and 0.1 μm Tetra-Speck microspheres were embedded for drift correction during dSTORM imaging.

### Single-molecule imaging with dSTORM

Single-molecule localisation microscopy technique dSTORM imaging was performed using a TIRF microscope (Nikon N-STORM 5.0) with a 100x oil immersion objective and multiple lasers. The imaging buffer

contained TN buffer (50 mM Tris-HCl pH 8.0, 10 mM NaCl), GLOX oxygen scavenger system [0.5 mg/ml glucose oxidase (Sigma-Aldrich, #G2133); 40 mg/ml catalase (Sigma-Aldrich, #C-100); and 10% w/v glucose], and 10 mM 2-aminoethanethiol (MEA; Sigma-Aldrich, #M6500). Image sequences for dSTORM were acquired on a total internal reflection fluorescence (TIRF) microscope (Nikon N-STORM 5.0) equipped with a 100x (1.49 NA) oil immersion objective and lasers (405 nm, 473 nm, 561 nm and 640 nm). Time series of 10,000 frames were acquired per sample, per channel (640 or 561 nm laser channel with continuous low-power 405 nm illumination) with an exposure time of 30 ms in TIRF mode. For dual-colour acquisition, the higher wavelength channel (640 nm laser for Alexa Fluor 647) was acquired first, followed by the channel with shorter wavelength (561 nm laser for Alexa Fluor 568) using a sCMOS camera (Hamamatsu Orca-Flash 4.0 V3). Image processing, including fiducial markers-based drift correction, two-channel alignment, and generation of x-y particle coordinates for each localisation, was carried out by NIS-Elements AR software (version 5.2).

### Cluster analysis of single-molecule images

The degree of TCR clustering was quantified by using a custom-built DBSCAN algorithm implemented in MATLAB. Pre-determined parameters included a minimum number of neighbours (3) and radius (20 nm). Clus-DoC analysis was performed to quantify the spatial distribution and colocalization of two proteins, in this case CD3ε and pCD3ζ. Density gradients were generated for each localisation and normalised. Spearman correlation was used to calculate the rank correlation coefficient. DoC scores ranged from +1 (colocalization) to −1 (segregation), with 0 indicating random distribution. The DoC threshold for colocalization was set to ≥ 0.1. One-way ANOVA with Tukey's multiple comparison was performed in GraphPad Prism v10. Statistical significance is denoted by asterisks with *$P \leq 0.05$, *$P \leq 0.01$, *$P \leq 0.001$ and *$P \leq 0.0001$; non-significant comparisons are not shown. Error bars denote S.E.M.

### Reporting summary

Further information on research design is available in the Nature Portfolio Reporting Summary linked to this article.

## Data availability

The γδ TCR 2-CD1d complex structure is available on the Protein Database under the accession code 9O4X [https://doi.org/10.2210/pdb9O4X/pdb]. Data generated for this study is present in the article and supplementary material. Source data are provided with this paper.

## Code availability

Code for the cluster analysis algorithm can be found at (https://github.com/PRNicovich/ClusDoC).

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

## Acknowledgements

This research was undertaken using the BioSAXS, SAXS/WAXS and MX2 beamlines at the Australian Synchrotron, part of the Australian Nuclear Science and Technology Organisation, and made use of the Australian Cancer Research Foundation detector. We thank Dr Ashish Sethi and the staff at the Australian Synchrotron for their assistance with data collection. We thank the staff from the Monash FlowCore, Monash Molecular Crystallisation Platform and Micromon facilities for their assistance. We thank Dr. Adam Uldrich for providing the 9C1 αβ TCR pMIG-II plasmid. M.T.R. was supported by an Australian Institute of Nuclear Science and Engineering Early Career Researcher Grant and a Faculty of Medicine, Nursing and Health Sciences Early Career Postdoctoral Fellowship. S.D.G. would like to thank the staff at the Centre for Dynamic Imaging at Walter and Elisa Hall Institute of Medical Research for providing access to the Single-Molecule Imaging microscope.

## Author contributions

M.T.R. - 1st author, collected, analysed data, wrote paper. S.D.G., single-molecule imaging analyses. C.L.S., M.P., and C-S.L. – undertook experiments/analysed experiments, revised the paper - J.R. and B.S.G. – co-supervised project, co-wrote and revised paper, J.R. – project funding.

## Competing interests

The authors declare no competing interests.
