## [Peer Review File · Nature Communications]

$\gamma\delta$ T cell receptor recognition of CD1d in a lipid-independent manner

Corresponding Author: Professor Jamie Rossjohn

Version 0:

Reviewer comments:

Reviewer #1

(Remarks to the Author)

In this manuscript, Rice et al. characterize a panel of previously identified $\gamma\delta$ T cell-derived TCRs that react with CD1d presenting the glycolipid antigen α -galactosylceramide. The authors demonstrate that some of these clones directly interact with the antigen within the groove of CD1d, while others interact only with the CD1d molecule itself, suggesting diverse modes of CD1d recognition by V δ 1-containing $\gamma\delta$ T cells. Through contrasting affinity measurements obtained using SPR and activation of $\gamma\delta$ TCR avatars in Jurkat cells, the authors further show that steady-state affinity measurements are not predictive of T cell activation capacity. They also reveal that allelic variants in the constant domain, leading to amino acid extensions in the $\gamma\delta$ TCR connecting peptide, affect signaling outcomes, with the C γ 2 allele showing reduced signaling and the C γ 3 allele leading to complete inhibition. Finally, they present the crystal structure of one such $\gamma\delta$ TCR in complex with CD1d presenting endogenous lipid derived from the mammalian expression system used, demonstrating that this TCR binds CD1d independently of the lipid antigen.

Comments: The authors choose to model sphingomyelin in the presenting cleft of CD1d produced by the mammalian expression system used here. While sphingomyelin is certainly a component of the CD1d-“endo” lipid repertoire, it is not the most abundant lipid species, with membrane phospholipids likely being more prevalent. Despite $\gamma\delta$ TCR 2 binding to CD1d independently of lipid antigens, does this suggest that there is still a certain level of selectivity for sphingomyelin over other self-antigens in the context of $\gamma\delta$ T cell recognition?

Reviewer #2

(Remarks to the Author)

The manuscript by Rice et al. presents a detailed structural and functional analysis of CD1d-restricted human V δ 1 $\gamma\delta$ TCRs. Using a powerful combination of surface plasmon resonance, SEC-SAXS, crystallography, mutagenesis, and single-molecule imaging, the authors dissect how $\gamma\delta$ TCRs engage CD1d in either lipid-dependent or lipid-independent fashions. The structure of a V δ 1V γ 5 TCR bound to CD1d-endo lipids reveals an antibody-like, lipid-independent binding mode, extending an emerging theme of unconventional $\gamma\delta$ TCR recognition that diverges from the classic TCR-MHC/CD1 co-recognition paradigm.

This is an ambitious and technically impressive study, which makes a substantial conceptual advance in our understanding of $\gamma\delta$ TCR biology. The findings will be of high interest to the immunology, T-cell biology, and structural immunology communities.

Major comments:

- The abstract and sections of the results are dense with technical detail (SAXS parameters, residue-level contacts). These obscure the main conceptual advance. The text would benefit from sharpening to emphasise the key message: $\gamma\delta$ TCRs can engage CD1d via lipid-independent binding, and ligand affinity alone is a poor predictor of functional outcome.
- Most activation data rely on Jurkat reporter cells. While highly informative, they do not fully recapitulate primary $\gamma\delta$ T-cell signalling. At minimum, the authors should explicitly acknowledge this limitation and, if possible, provide some corroboration in primary V δ 1+ T cells.
- The terms “lipid-independent” and “auto-reactive” are used interchangeably. These should be defined clearly and used consistently, as the distinction is mechanistically important.

Minor comments

- Replace awkward phrasing such as “rarefied within peripheral blood” with clearer immunological terms (e.g. “present at low frequency”).
- Figures could be strengthened by schematic cartoons illustrating lipid-dependent vs lipid-independent recognition modes and V γ 5 TCR dimerisation.
- The AML/MM link is interesting but speculative. It should be presented cautiously unless backed by primary patient data.

Reviewer #3

(Remarks to the Author)

Rice et al characterize the antigen specificities, binding modes, and activation determinants for a panel of previously identified CD1d-specific human Vd1 gamma-delta TCRs by SPR and SAXS using recombinant proteins and by T cell activation and TCR clustering studies using Jurkat transfectants. They determine the X-ray crystal structure for one lipid-independent gdTCR, and characterize the role of allelic variation the constant domain C γ connecting peptide region was studied for one of the TCRs.

For the several gdTCRs studied, CD1d is recognized independently of bound lipid, unlike the canonical iNKT abTCR recognizing CD1d-aGalSer which is highly dependent on TCR-lipid contacts. This antigen-independent (“autoreactive”) recognition is similar in some ways to that previously observed for other gdTCR that recognize CD1a or CD1b but in this case the binding mode is like that for the lipid-dependent iNKT abTCR, as compared to the wildly divergent binding modes observed for the gdTCRs. Here the overall MHC-TCR orientation and CDR usage are relatively conventional, with a small shift to the A' side of the binding site away from the exposed (endogenous) lipid head group. The CDR3 γ is relatively short relative to iNKTTCR, and the authors argue that this contributes to lipid-independent recognition of CD1d, along with the robust participation of other CDR loops and the FW γ (FW4 γ ?) in hydrophobic interactions with CD1d exposed Trp and other residues. These results highlight the complex and variable role that CD1d-bound lipid can play in TCR recognition. The issue of domain flexibility in gdTCR recognition recently has been highlighted in several studies. Here the authors studied in detail the effect of constant domain C γ allelic variants with varying length insertion into the connecting region, and found for the longest variants relatively tight binding and TCR clustering can occur without substantial activation. These results contribute to developing understanding of the requirements for gdT cell activation and how these differs from conventional abT cell activation requirements. Overall this manuscript deepens our understanding of molecular aspects of gdTCR recognition of CD1, although distinct immunological roles of for antigen-specific and autoreactive recognition are still elusive.

There are few things in the manuscript that could be clarified

In Fig 1b, the ab initio SAXS models would be easier to interpret if the TCR and CD1d components had domain/chain labels. In particular the CD1d components of the gdTCR7 and gdTCR8 models are difficult to find – or are these just gdTCR dimers without CD1d?

Also in Fig 1b, the last ab initio model is a bit confusing, since all the others line up with the TCR cartoon and TCR-CD1d SPR data in panel A. The last model shown only CD1d SAXS, but I was expecting data for abNKT TCR-CD1d complex for comparison with the gdTCR-CD1d complexes. Are SAXS data available for the abNKT TCR – CD1d complex? If not perhaps the CD1d-only SAXS data and model could be relocated or set off somehow.

In Suppl Table 2, there should be an additional column providing the observed stoichiometry, and one with calculated molecular mass for that stoichiometry for comparison with experimental mass in the fifth column.

Were the C γ exon 3 allelic variants differentially associated with the particular TCR sequences used here, or was the 9C2 TCR just used for convenience to examine the general phenomenon?

Version 1:

Reviewer comments:

Reviewer #1

(Remarks to the Author)

No Further comment

Reviewer #2

(Remarks to the Author)

Thank you for your thorough revisions. The authors have addressed all the previous concerns comprehensively, and the manuscript has improved substantially in clarity and rigour. I am satisfied with the changes made and have no further major comments. I believe the paper is now suitable for publication in its current form.

Reviewer #3

(Remarks to the Author)

The revised manuscript addressed the comments and concerns expressed in my review of the original manuscript.

REVIEWER COMMENTS – Point by Point Response

Reviewer #1:

1) The authors choose to model sphingomyelin in the presenting cleft of CD1d produced by the mammalian expression system used here. While sphingomyelin is certainly a component of the CD1d-“endo” lipid repertoire, it is not the most abundant lipid species, with membrane phospholipids likely being more prevalent. Despite $\gamma\delta$ TCR 2 binding to CD1d independently of lipid antigens, does this suggest that there is still a certain level of selectivity for sphingomyelin over other self-antigens in the context of $\gamma\delta$ T cell recognition?

We thank reviewer #1 for highlighting a point of potential confusion. As we show in the crystal structure, $\gamma\delta$ TCR-2 does not contact the lipid antigen. Although we observed density for a lipid antigen within the CD1d groove, we are indeed unable to precisely determine the lipid species present as it likely contained a mixture of lipid antigens.

However, informed by our previous CD1d-‘endo’ crystal structures and complementary mass-spectrometry analysis (Lameris et al, 2023, Journal for ImmunoTherapy of Cancer & Huang et al 2023, Cell) and by others (Haig et al 2011, Journal of Biological Chemistry), we modelled sphingomyelin C24:1, as the density indicated the presence of a phospholipid antigen. However, as Reviewer #1 suggests we cannot exclude CD1d-‘endo’ containing other phospholipids. We have updated the results section accordingly to clarify this – Page 8, Line 262:

“Although we determined the structure of the $\gamma\delta$ TCR-CD1d complex with CD1d carrying a mixture of endogenous lipids from the mammalian expression system, density was observed within the antigen presenting cleft of CD1d indicating the abundance of one class of lipid antigen. Here, we modelled sphingomyelin which has previously been identified as a component of the CD1d-‘endo’ lipid repertoire (**Supplementary Figure 6A and B**)^{34,35}. Despite sphingomyelin being modelled into the crystal structure, we cannot exclude the possibility our crystal structure likely contains a mixture of phospholipid antigens.”

Reviewer #2:

We thank the Reviewer for their positive appraisal of our manuscript and for stating ‘**This is an ambitious and technically impressive study, which makes a substantial conceptual advance in our understanding of $\gamma\delta$ TCR biology. The findings will be of high interest to the immunology, T-cell biology, and structural immunology communities.**’

1) The abstract and sections of the results are dense with technical detail (SAXS parameters, residue-level contacts). These obscure the main conceptual advance. The text would benefit from sharpening to emphasise the key message: $\gamma\delta$ TCRs can engage CD1d via lipid-independent binding, and ligand affinity alone is a poor predictor of functional outcome.

We have updated the abstract to simplify our activation data and focus on the key findings – Page 2, Line 29:

“In functional assays, CD1d $\gamma\delta$ TCR affinity was a poor predictor of $\gamma\delta$ T cell line activation. Moreover, CD1d presenting endogenous lipids was sufficient to stimulate T cell activation and induce $\gamma\delta$ TCR-CD3 clustering and phosphorylation in a dose dependent manner”

We have also updated the SAXS analysis to remove unnecessary technical detail – Page 4, Line 130:

“The scattering profiles for the V γ 2 $\gamma\delta$ TCR3 and V γ 9 $\gamma\delta$ TCR6 were consistent with a globular protein. Compared to $\gamma\delta$ TCRs 3 and 6, the $\gamma\delta$ TCRs 2 and 8 scattering profiles were indicative of larger protein samples that suggested TCR dimerisation, with an oligomeric status of 1.8 and 1.5 respectively (**Supplementary Table 2, Supplementary Figure 2**). This appears to be a conserved feature with V γ 5 TCRs such as $\gamma\delta$ TCRs 2 and 8, corroborating previous SAXS, X-ray crystallography and cryo-EM experiments^{19, 23, 27, 28}. Evaluation of the scattering profiles by comparison to known macromolecular structures of a V γ 5 TCR and V γ 5 TCR dimer confirmed that the $\gamma\delta$ TCRs 2 and 8 formed a dimer in solution (**Supplementary Figure 3**). SAXS analyses of CD1d revealed scattering consistent with a globular monomer in solution.

Upon complexation with CD1d, $\gamma\delta$ TCRs 2, 3 and 6 shifted on SEC, indicative of complex formation (**Supplementary Figure 3C**). We next generated, aligned and averaged multiple *ab initio* reconstructions of the $\gamma\delta$ TCR 2, 3 and 6-CD1d-‘endo’ complex samples.”

We have also updated the crystal structure analysis to remove unnecessary detail and appeal to a broader immunological audience – Page 9, Line 301:

“The total buried surface area (BSA) of the TCR-CD1d interface was $\approx 1860\text{\AA}^2$ (**Figure 6A**). The TCR chains contributed asymmetrically to the interface with $\approx 59\%$ of the interface mediated by TCR- γ and $\approx 41\%$ by TCR- δ (**Figure 6A**). This contrasts with the emerging trend of V δ 2- $\gamma\delta$ TCR-ligand recognition being dominated by the TCR- δ chain, in particular CDR3 δ ^{36, 37}.

Notably, FW γ contributed to this interface with several polar interactions to CD1d (**Figure 6B**)(**Supplementary Table 4**). This served to stabilise CDR1 γ and CDR2 γ , which were positioned either side of the CD1d- α 1 helices and made a number of key contacts to CD1d (**Figure 6B**).

A key determinant in lipid binding for the 9C2 V δ 1V γ 5 TCR was CDR3 γ , namely Arg103 γ and Tyr11 γ that directly contact the head-group of α -GalCer. Instead, the CDR3 γ of $\gamma\delta$ TCR 2 did not contact the lipid antigen (**Figure 6C**), providing the molecular basis for the lipid-independent binding identified in the SPR experiments. Rather CDR3 γ solely contacted the CD1d molecule itself (**Figure 6C**). The 9C2 TCR and $\gamma\delta$ TCR2 possess different CDR3 γ loop lengths, 18 versus 13 amino acids, which may account for the differences identified in biochemical and structural based observations. The shorter CDR3 γ loop of $\gamma\delta$ TCR 2 leads to a rearrangement of the TCR-CD1d interface compared

to the 9C2 TCR, shifting the CDR3 γ/δ loops ~ 8 Å towards the A' of CD1d, preventing direction recognition of the presented lipid antigen.

TCR- δ chain binding was predominately mediated by the CDR3 δ loop. CDR2 δ did not contact CD1d while CDR1 δ made minimal contacts, a feature not seen in other CD1d- $\gamma\delta$ TCR complexes that relied heavily on CDR1 δ for CD1d engagement^{18,19}. Central to this recognition was Trp99 δ , which wedged in-between the α helices of CD1d, forming an extensive network of interactions with CD1d (**Figure 6D**). Similar to CDR3 γ , the CDR3 δ loop of $\gamma\delta$ TCR-2 is shorter compared to 9C2, 12 vs 16 amino acids. Consequently, neither CDR3 γ or CDR3 δ made contacts to the presented lipid antigen providing structural evidence for lipid independent CD1d binding by a $\gamma\delta$ TCR.”

2) Most activation data rely on Jurkat reporter cells. While highly informative, they do not fully recapitulate primary $\gamma\delta$ T-cell signalling. At minimum, the authors should explicitly acknowledge this limitation and, if possible, provide some corroboration in primary V δ 1+ T cells.

Please note that our prior Uldrich et al study (Nature Immunology 2013), investigated primary V δ 1+ T cell activation data. Please see Figure 3C from this study, which we also show here for the reviewer’s convenience.

Figure 3C from Uldrich et al 2013. CD69 mean fluorescence intensity (MFI; arbitrary units) of gated V δ 1⁺ $\gamma\delta$ TCR⁺ CD1d- α -GalCer tetramer⁺ cells (left) and V δ 1⁻ CD1d- α -GalCer tetramer⁺ cells (‘type I NKT’, right), derived from in vitro-expanded CD1d- α -GalCer tetramer-enriched PBMCs that were cocultured with C1R parental or C1R CD1d^{hi} APCs with or without α -GalCer for 6 h. Data represent 6 separate donors, screened in one experiment.

We have updated the introduction to highlight our previous activation and tetramer staining experiments performed on primary $\gamma\delta$ T cells from multiple healthy donors, from which our $\gamma\delta$ TCR panel was sourced – Page 3, Line 91.

“Previous tetramer staining and activation experiments on primary V δ 1⁺ $\gamma\delta$ ⁺ CD1d- α -GalCer⁺ cells, suggested donor specific patterns of lipid -independent and -dependent cell staining and activation¹⁹. We characterised a panel of these previously identified CD1d- α -GalCer reactive V δ 1⁺ T cells to determine the extent to which V δ 1⁺ CD1d binding was lipid antigen dependant¹⁹.”

We have also updated the discussion to reflect the limitations in our cellular activation experiments – Page 11, Line 347:

“A limitation of this study is the use of Jurkat T cells to investigate $\gamma\delta$ T cell activation raising questions on whether our conclusions extend to primary $\gamma\delta$ T cells. We have previously demonstrated that PBMC derived CD1d-restricted $\gamma\delta$ T cells activate in response to cells over-expressing CD1d in the absence of lipid pulsing¹⁹. This alludes to CD1d-restricted primary $\gamma\delta$ T cell activation also being dependent on ligand availability.”

3) The terms “lipid-independent” and “auto-reactive” are used interchangeably. These should be defined clearly and used consistently, as the distinction is mechanistically important.

We agree with Reviewer #2 that the interchangeable use of different immunological terms may be misleading to the reader. We have updated the manuscript accordingly removing the use of “auto-reactive” and instead using “lipid-independent”.

4) Replace awkward phrasing such as “rarefied within peripheral blood” with clearer immunological terms (e.g. “present at low frequency”).

We have altered the term “rarefied” to “present at low frequency”.

5) Figures could be strengthened by schematic cartoons illustrating lipid-dependent vs lipid-independent recognition modes and V γ 5 TCR dimerisation.

We thank Reviewer #2 for this suggestion. We have updated Figure 1A to include cartoon schematics depicting the observed lipid-dependent vs lipid-independent recognition modes in our SPR experiments. Further, we have updated Figure 1B to include a cartoon schematic depicting TCR dimerization in our SAXS analysis.

6) The AML/MM link is interesting but speculative. It should be presented cautiously unless backed by primary patient data.

We have now removed this tangential discussion point in the revision.

Reviewer #3:

We thank the Reviewer for their detailed review of our manuscript and for stating “Overall this manuscript deepens our understanding of molecular aspects of gdTCR recognition of CD1”.

1) In Fig 1b, the ab initio SAXS models would be easier to interpret if the TCR and CD1d components had domain/chain labels. In particular the CD1d components of the gdTCR7 and gdTCR8 models are difficult to find – or are these just gdTCR dimers without CD1d?

Thank you for this suggestion. Accordingly, we have now updated Figure 1B to include chain labels for the $\gamma\delta$ TCR and CD1d components. Further, we have included a cartoon schematic for $\gamma\delta$ TCRs 7 and 8 illustrating TCR dimerization via the V γ 5 chains.

2) Also in Fig 1b, the last ab initio model is a bit confusing, since all the others line up with the TCR cartoon and TCR-CD1d SPR data in panel A. The last model shown only CD1d SAXS, but I was expecting data for abNKT TCR-CD1d complex for comparison with the gdTCR-CD1d complexes. Are SAXS data available for the abNKT TCR – CD1d complex? If not perhaps the CD1d-only SAXS data and model could be relocated or set off somehow.

We agree with Reviewer #3 that SAXS comparison between $\alpha\beta$ NKT-CD1d and $\gamma\delta$ TCR-CD1d would have been informative. We had attempted across two separate SEC-SAXS experiments to generate this data, however we were unable to generate ab initio models of $\alpha\beta$ NKT-CD1d complexes, instead generating models of a mixture of both individual components. We thank Reviewer #3 for highlighting a potential point of confusion and subsequently we have relocated the CD1d SEC-SAXS *ab initio* model to Supplementary Figure 2.

3) In Suppl Table 2, there should be an additional column providing the observed stoichiometry, and one with calculated molecular mass for that stoichiometry for comparison with experimental mass in the fifth column.

We thank the Reviewer for this suggestion, accordingly we have updated Supplementary Table 2.

4) Were the C γ exon 3 allelic variants differentially associated with the particular TCR sequences used here, or was the 9C2 TCR just used for convenience to examine the general phenomenon?

We selected the 9C2 $\gamma\delta$ TCR as the background for the C γ exon alleles, as this is a well characterised TCR-ligand system and had been the focus of recent Cryo-EM and functional analyses by Xin et al (Nature) and Hoque et al (Nature Communications). Unfortunately, we did not possess data on which C γ alleles of our $\gamma\delta$ TCR panel expressed. We have updated the Results section to highlight why we selected the 9C2 TCR – Page 7, Line 238.

“We investigated whether the exon 2 triplication event, C γ 3, had a further impact on $\gamma\delta$ TCR signalling outcomes, using the previously well characterised CD1d restricted 9C2 $\gamma\delta$ TCR^{19, 27}.”